# Intratumoral Adipocyte-High Breast Cancer Enrich for Metastatic and Inflammation-Related Pathways but Associated with Less Cancer Cell Proliferation

**DOI:** 10.3390/ijms21165744

**Published:** 2020-08-11

**Authors:** Yoshihisa Tokumaru, Masanori Oshi, Eriko Katsuta, Li Yan, Jing Li Huang, Masayuki Nagahashi, Nobuhisa Matsuhashi, Manabu Futamura, Kazuhiro Yoshida, Kazuaki Takabe

**Affiliations:** 1Breast Surgery, Department of Surgical Oncology, Roswell Park Comprehensive Cancer Center, Buffalo, NY 14263, USA; yoshitoku1090@gmail.com (Y.T.); masanori.oshi@roswellpark.org (M.O.); eriko.katsuta@roswellpark.org (E.K.); jingli.huang@roswellpark.org (J.L.H.); 2Department of Surgical Oncology, Graduate School of Medicine, Gifu University, 1-1 Yanagido, Gifu 501-1194, Japan; nobuhisa@gifu-u.ac.jp (N.M.); mfutamur@gifu-u.ac.jp (M.F.); kyoshida@gifu-u.ac.jp (K.Y.); 3Department of Gastroenterological Surgery, Yokohama City University Graduate School of Medicine, Yokohama 236-0004, Japan; 4Department of Biostatistics & Bioinformatics, Roswell Park Comprehensive Cancer Center, Buffalo, NY 14263, USA; li.yan@roswellpark.org; 5Department of Surgery, Niigata University Graduate School of Medical and Dental Sciences, Niigata 951-8510, Japan; masanagahashi@gmail.com; 6Department of Surgery, University at Buffalo Jacobs School of Medicine and Biomedical Sciences, The State University of New York, Buffalo, NY 14263, USA; 7Department of Breast Oncology and Surgery, Tokyo Medical University, 6-7-1 Nishishinjuku, Shinjuku, Tokyo 160-8402, Japan; 8Department of Breast Surgery, Fukushima Medical University School of Medicine, Fukushima 960-1295, Japan

**Keywords:** tumor-associated adipocyte, breast cancer, tumor immune microenvironment, xCell, adipokines, gene set enrichment analysis, TCGA, METABRIC

## Abstract

Cancer-associated adipocytes are known to cause inflammation, leading to cancer progression and metastasis. The clinicopathological and transcriptomic data from 2256 patients with breast cancer were obtained based on three cohorts: The Cancer Genome Atlas (TCGA), GSE25066, and a study by Yau et al. For the current study, we defined the adipocyte, which is calculated by utilizing a computational algorithm, xCell, as “intratumoral adipocyte”. These intratumoral adipocytes appropriately reflected mature adipocytes in a bulk tumor. The amount of intratumoral adipocytes demonstrated no relationship with survival. Intratumoral adipocyte-high tumors significantly enriched for metastasis and inflammation-related gene sets and are associated with a favorable tumor immune microenvironment, especially in the ER+/HER2- subtype. On the other hand, intratumoral adipocyte-low tumors significantly enriched for cell cycle and cell proliferation-related gene sets. Correspondingly, intratumoral adipocyte-low tumors are associated with advanced pathological grades and inversely correlated with *MKI67* expression. In conclusion, a high amount of intratumoral adipocytes in breast cancer was associated with inflammation, metastatic pathways, cancer stemness, and favorable tumor immune microenvironment. However, a low amount of adipocytes was associated with a highly proliferative tumor in ER-positive breast cancer. This cancer biology may explain the reason why patient survival did not differ by the amount of adipocytes.

## 1. Introduction

Breast cancer is one of the leading causes of cancer-related deaths in developed countries, including the United States [1,2]. Obesity is one of the known risk factors linked to the development and progression of breast cancer, particularly in the estrogen receptor-positive (ER+) subtype [3,4]. Excess adipose tissue secretes inflammatory cytokines and mediators [3,5]. Our group has indeed previously reported that a lipid mediator, sphingosine-1-phosphate, is more abundantly produced in the obese body and links obesity, inflammation, breast cancer metastasis [6], and treatment resistance [7]. In addition to adipose tissue, the role of adipocytes in the tumor microenvironment has lately drawn considerable attention [8].

Numerous investigators, including our group, have reported that not only cancer cells but also the tumor microenvironment plays a critical role in cancer progression and metastasis, as well as therapeutic responses [9,10]. Adipocytes, together with vascular endothelial cells [6,11,12], fibroblasts [13,14], and immune cells [15,16,17,18,19] constitute the tumor microenvironment [20,21,22]. Cancer-associated adipocytes interact with cancer cells and secrete the inflammatory cytokines, such as IL-6 and TNF-α, which contribute to pro-cancer inflammation that is known to aggravate cancer progression [23,24,25,26,27]. IL-6 promotes angiogenesis, invasion, and metastasis by activating the JAK/STAT3 pathway [28,29]. TNF-α promotes the invasion and metastasis of breast cancer by activating epithelial–mesenchymal transition (EMT) signaling [30,31,32]. In addition, cancer-associated adipocytes secrete chemokines such as Chemokine (C–C motif) ligand 2 (CCL2), which activates NOTCH1 signaling pathway and induces the stemness of the cancer cells [22,33]. Leptin, which is also produced by cancer-associated adipocytes, collaborates with IL-1, and promotes angiogenesis by upregulating expression of VEGF and VEGF receptors [22,34,35,36].

Although pre-clinical studies have shown how adipocytes may affect the progression of cancer, little is known about the clinical relevance of the amount of adipocytes in breast cancer. Recently, our group has been assessing the clinical relevance of the tumor microenvironment utilizing in silico approaches [10,14,17,18,37,38,39,40,41].

We demonstrated that factors affecting favorable tumor immunity may lead to better clinical outcomes. For instance, we demonstrated that pancreatic cancer with a higher amount of fibroblasts is associated with anti-cancer immunity and a higher rate of complete resection without residual tumor (R0) [14]. In addition, we reported that breast cancers with high expression of microRNA-143 were associated with a favorable tumor immune microenvironment and led to better survival [18]. In a similar fashion, we estimated the amount of adipocytes utilizing xCell, a bioinformatics algorithm that defined adipocytes by the expression pattern of 21 genes, as reported by Aran D et al. (Appendix A) [42].

Here, we hypothesized that breast cancer with a high amount of intratumoral adipocytes is associated with inflammation, metastasis, and worse survival. The aim of this study was to test this hypothesis utilizing in silico analysis of publicly available large databases.

## 2. Results

### 2.1. Transcriptionally Defined Adipocytes Significantly Correlated with Markers of Mature adipocytes

First, we assessed whether transcriptionally defined adipocytes appropriately reflected the characteristics of differentiated adipocytes by analyzing the correlation with their key markers: Adiponectin, leptin, lipoprotein lipase, and perilipin 1 (gene names are *ADIPOQ, LEP, LPL, and PLIN1,* respectively). Based on the TCGA cohort, transcriptionally defined adipocytes were strongly correlated with adiponectin, leptin, lipoprotein lipase, and perilipin 1 (Figure 1A; *r* = 0.867, *r* = 0.794, *r* = 0.672, and *r* = 0.875, respectively). We used two other cohorts to validate these results. Adipocytes were either moderately or strongly correlated with adiponectin, leptin, lipoprotein lipase, and perilipin 1 in the GSE25066 cohort (Figure 1B; *r* = 0.718, *r* = 0.472, *r* = 0.479, and *r* = 0.718, respectively) and the cohort of Yau et al. (Figure 1C; *r* = 0.758, *r* = 0.477, *r* = 0.615). No data for perilipin 1 was available in the latter cohort due to a lack of probe for *PLIN1*. These results implicated that transcriptionally defined adipocytes reasonably reflected mature adipocytes in a bulk tumor.

### 2.2. Breast Cancer Has High Level of Intratumoral Adipocytes Compared to the Other Cancers

Our methodology to analyze gene signatures to quantify adipocytes allowed us to measure the amount of adipocytes in any gene expression dataset, including the entire TCGA cohort. Since samples cataloged in TCGA were collected from a cancer cell-rich portion of the bulk tumors, we specifically termed our findings as “intratumoral adipocytes” to differentiate from infiltrating adipocytes in the peritumoral region. Utilizing our method, we compared the amount of intratumoral adipocyte among the different types of cancer in TCGA. Breast cancer (BRCA) possessed the second highest amount of adipocytes among the various cancers in TCGA, trailing only to hepatocellular carcinoma (LIHC) (Figure 1D). The other cancers with a relatively high amount of adipocytes included brain cancer (LGG, GBM), kidney cancer (KIRC, KICH, KIRP), mesothelioma (MESO), and cholangiocarcinoma (CHOL). On the contrary, almost no adipocytes were detected in melanoma (SLCM, UVM) or leukemia (LAML).

### 2.3. Intratumoral Adipocyte-High Tumors Were Not Associated with Worse Survival in Any of the Subtypes, nor with Pathological Complete Response (pCR) to Neoadjuvant Chemotherapy (NAC)

Given the detrimental role of cancer-associated adipocytes [26], we investigated whether the amount of intratumoral adipocyte was associated with the clinical outcomes in breast cancer. We divided the patients into high and low groups by the amount of adipocytes using a median cutoff. There was no statistically significant association between the amount of intratumoral adipocyte and patient survival, although intratumoral adipocyte-high tumors demonstrated a tendency for better overall survival (OS) in the whole TCGA cohort and the estrogen receptor positive/HER2 negative (ER+/HER2-) subtype (Figure 2; *p* = 0.085 and *p* = 0.088, respectively). There was no association between intratumoral adipocyte-high tumors and survival in either the HER2 positive (HER2+) or the triple negative (TNBC) subtype (Figure 2).

A previous in vitro study reported that the co-culture of adipocyte with human and murine cell lines promoted the resistance to doxorubicin, paclitaxel, and 5-fluorouracil (5-FU) independently from the subtypes [43]. Thus, we expected that intratumoral adipocyte-high tumors were associated with decreased pathological response (pCR) rate. We analyzed three independent neoadjuvant chemotherapy (NAC) cohorts that underwent different chemotherapy regimens; GSE25066 (*n* = 508; taxane and anthracycline) [44], GSE32646 (*n* = 115; 5-FU, epirubicin, cyclophosphamide, and paclitaxel) [45], and GSE20194 (*n* = 278; paclitaxel, 5-FU, cyclophosphamide and doxorubicin) [46]. Unexpectedly, intratumoral adipocyte-high tumors were not associated with decreased pCR rate in both ER+/HER2- and TNBC subtypes in any of NAC cohorts we analyzed (Appendix A).

### 2.4. Intratumoral Adipocyte-High Tumors were not Associated with Advanced Cancer Staging

Cancer-associated adipocytes are known to promote tumor growth, invasion, and migration of breast cancer cells in vitro [25]. To this end, we hypothesized that tumors that were high in intratumoral adipocytes would be associated with the advanced American Joint Commission on Cancer (AJCC) clinical cancer staging in the TCGA cohort. Contrary to our expectation, less aggressive stage 1 tumors tended to associate with increased intratumoral adipocytes compared to tumors of other stages in the whole cohort and in any of the subtypes; however, none of these associations were statistically significant (Appendix A; all *p* > 0.05). 

### 2.5. Intratumoral Adipocyte-High Tumors Enriched for Inflammation, Metastasis, and Immune Response-Related Gene Sets

In order to examine whether intratumoral adipocytes demonstrated similar biological functions as previously reported in vitro [25], a gene set enrichment assay (GSEA) was performed on the TCGA cohort. In the whole cohort and in both ER+/HER2- and TNBC subtypes, intratumoral adipocyte-high tumors enriched for inflammation-related gene sets: TNF-α signaling via NFκB, IL-6/JAK/STAT3 signaling, and inflammatory response (Figure 3A, Appendix A). This was concordant with previous reports [25,26]. The aforementioned gene sets were not enriched in the GSE25066 cohort, which had fewer patients compared with TCGA. We also found intratumoral adipocyte-high tumors enriched for epithelial–mesenchymal transition (EMT), TGFβ signaling, angiogenesis, WNTβ catenin signaling, and Hedgehog signaling in the whole cohort and the ER+/HER2 subtype in TCGA (Figure 3B, Appendix A). These gene sets reflected on mechanisms of metastasis and cancer stemness. These were also enriched in intratumoral adipocyte-high tumors in the GSE25066 cohort. Only in the ER+/HER2 subtype was intratumoral adipocytes-high tumors enriched for Notch signaling. In the TNBC subtype, only TGFβ signaling was enriched among the metastasis and cancer stemness-related gene sets (Appendix A). On the other hand, intratumoral adipocyte-high tumors enriched for gene sets related to immune response such as IL2 signaling and IFN-γ only in ER+/HER2- subtype in TCGA cohort (Figure 3C, Appendix A). In the GSE25066 cohort, only IL2 STAT5 signaling was enriched among the immune response gene sets. These results suggest that intratumoral adipocyte-high tumors were associated with inflammation, metastasis, and immune-related gene sets, especially in the ER+/HER2- subtype.

### 2.6. Intratumoral Adipocyte-High Tumors Were Associated with Favorable Tumor Immune Microenvironment in ER+/HER2- Subtype

Given the data from the gene set enrichment analysis, we further explored the infiltration of immune cells within the tumor microenvironment using the whole TCGA breast cancer cohort. Regarding favorable immune cells, intratumoral adipocyte high tumors were associated with a high proportion of dendritic cells (DC), but with less with type 1 helper T cells (Th1) and M1 macrophages (M1) in the whole cohort (Figure 4A; all *p* < 0.001). In terms of unfavorable immune cells, less type 2 helper T cells (Th2), regulatory T cells (Treg), and M2 macrophages (M2) were found in intratumoral adipocyte-high tumors, compared to adipocyte-low tumors (Figure 4B; all *p* < 0.001). A similar pattern of results was found in the ER+/HER2- subtype, whereas only Th1 and Treg were associated with intratumoral adipocyte-high tumors in the TNBC subtype (Figure 4A,B). The cytolytic activity score (CYT) reflects the cytolytic activity of all the immune cells in the bulk tumor. High CYT was associated with intratumoral adipocyte-high tumors in the whole cohort and in the ER+/HER2- subtype, but not in TNBC subtype (Figure 4C). These results indicated that adipocyte-high tumors were associated with overall favorable tumor immune microenvironment in ER+/HER2- subtype, but not in TNBC.

### 2.7. Intratumoral Adipocyte-Low Tumor Enriched Cell Cycle- and Cell Proliferation-Related Gene Sets

We further investigated gene sets that may be associated with intratumoral adipocyte-low tumors, given that tumors with high adipocytes were enriched for metastasis and inflammation-related gene sets but did not translate to a survival difference. We performed GSEA on intratumoral adipocyte-low tumors in the TCGA and GSE25066 cohorts. We found that intratumoral adipocyte-low tumors significantly enriched for cell cycle and cell proliferation gene sets such as E2F targets, G2M checkpoint, MYC targets V1, and MYC targets V2 in the whole breast cancer cohort of TCGA. Similar results were found in patients with the ER+/HER2 subtype in the GSE25066 cohort (Figure 5, Appendix A). These results demonstrate that intratumoral adipocyte-low tumors are associated with increased cell cycle and cell proliferation.

### 2.8. Adipocyte Low Tumor Correlated with High MKI67 Expression, and Was Associated with Advanced Nottingham Pathological Grade

Given that intratumoral adipocyte-low tumor strongly enriched most of the cell proliferation gene sets, we hypothesized that intratumoral adipocyte low tumors were a clinically highly proliferative cancer. We examined the association between the amount of intratumoral adipocyte and *MKI67* expression, the most commonly used marker of cancer cell proliferation in the clinical setting [37,47,48]. As we expected, intratumoral adipocyte low tumors demonstrated significantly higher *MKI67* expression consistently across all three cohorts TCGA, GSE25066, and the cohort from Yau et al. (Figure 6A, *p* < 0.001, *p* < 0.001, and *p* < 0.001, respectively). Furthermore, the amount of intratumoral adipocytes was weakly inversely correlated with *MKI67* expression in TCGA (Figure 6B; *r* = −0.235, *p* < 0.01). This result was validated with two other cohorts, GSE25066 and the cohort reported by Yau et al. (Figure 6B; *r* = −2.09 and *r* = −0.322, respectively).

In clinical practice, cell proliferation was assessed morphologically by Nottingham pathological grade [18]. We found that the intratumoral adipocyte-low tumors were associated with advanced grade, grade 2 or 3, compared with grade 1 in the whole cohort and in the ER+/HER2- subtype (Figure 6C; *p* < 0.001 and *p* < 0.001 respectively). This result was also true in GSE25066 cohort (Figure 6C).

## 3. Discussion

In the current study, transcriptionally defined adipocytes reasonably correlated with previously reported markers of mature adipocytes. Breast cancer contained the second highest amount of intratumoral adipocytes among the cancers listed in TCGA. There was no survival difference nor pCR rate after NAC by the amount of intratumoral adipocytes in any of the subtypes in the TCGA cohort, nor in any of the NAC cohort. Intratumoral adipocyte-high tumors significantly enriched for inflammation and metastasis-related gene sets as well as favorable tumor immune microenvironment, particularly in the ER+/HER2- subtype from TCGA and in the GSE25066 cohort. On the other hand, intratumoral adipocyte-low tumors significantly enriched for cell cycle and cell proliferation-related gene sets in all subtypes in both the TCGA and GSE25066 cohorts. Intratumoral adipocyte-low tumors were significantly associated with a higher *MKI67* expression and demonstrated an inverse correlation. Intratumoral adipocyte-low tumors were associated with advanced pathological grades, which is a clinical parameter of aggressive, highly proliferative breast cancer.

We have found that intratumoral adipocyte-high breast cancer enriched for the inflammation, metastasis, and cancer stemness related gene sets, which is in agreement with the previous reports [25,49,50,51,52]. This was more significant with the ER+/HER2- subtype compared with TNBC. Manabe et al. reported that the co-culture of mature adipocyte and ER+ breast cancer cells, such as MCF7, promoted the growth of these cancer cells [52]. In addition, a study by Lee et al. demonstrated that co-culture of adipocytes with cancer cells induced the upregulation of EMT-related genes, which led to the promotion of cancer cell invasion and migration [51]. Our study of clinical samples from large databases reflected an association with mechanisms of EMT that has been suggested by these in vitro experiments. Furthermore, intratumoral adipocyte-high tumors enriched for the gene sets associated with metastasis, such as TGFβ Signaling, Notch Signaling, Angiogenesis, WNTβ Catenin Signaling, and Hedgehog Signaling. This was more evident with the ER+/HER2- subtype compared with TNBC, which indicates that the role of adipocytes is more significant with the ER+/HER2- subtype. Our result is in perfect alignment with the previous studies that leptin, one of the well-known adipokines which enhances the cell proliferation and metastasis [53,54], and leptin receptor signaling crosstalk with ER signal pathway in ER+ subtypes but not in ER- subtypes [55,56].

Intratumoral adipocyte-high tumors are also enriched for inflammation-related gene sets, such as TNF-α signaling via NFκB, IL-6/JAK/STAT3 signaling, and inflammatory response in the TCGA cohort. Studies have shown the detrimental role of inflammatory cytokines, TNF-α and IL-6, in the promotion of cancer progression [6,7,12,17,40]. It has been reported that cancer associated adipocytes secrete IL-6 and TNF-α [25,26]. IL-6 activates the JAK/STAT3 pathway and promotes angiogenesis, invasion, metastasis [29,57], and TNF-α activate EMT and promote cancer progression [32]. Our result is consistent with these previous studies that a high amount of intratumoral adipokines associates with inflammation that aggravates cancer in the clinical setting.

Intratumoral adipocyte-high tumors also enriched for gene sets related to immune response such as IL2 signaling and IFN-γ and are associated with a smaller number of unfavorable immune cells such as Th2, Treg, and M2 macrophages, as well as with higher CYT. Overall, these associations suggest a favorable tumor immune microenvironment, especially in the ER+/HER2- subtype. Intratumoral adipocytes may contribute to breast cancer through inflammation, metastasis, and cancer stemness, but also associated with a favorable immune response that slows the cancer aggressiveness down, particularly in ER+/HER2- subtype.

It was surprising that the survival outcome did not differ between intratumoral adipocyte-high and low tumors in all the subtypes analyzed in the current study, especially when we found that intratumoral adipocyte associated with inflammation, metastasis, and cancer stemness, which have been previously reported to aggravate cancer [25,26,29]. Indeed, while the favorable immune response was associated with adipocyte-high tumors, there was no strong association with anti-cancer immune cell infiltration. Therefore, it is unlikely that favorable immune cell accumulation cancels out the detrimental effect of adipocytes. Conversely, we found was that intratumoral adipocyte-low tumors significantly enriched for four out of five designated hallmark cell proliferation gene sets (Figure 5), suggesting that they are extremely highly proliferative cancer. It is possible that the dense cellularity of highly proliferative cancer may not allow much space for adipocytes to infiltrate by physical pressure. Indeed, adipocytes are seldomly observed in the histological section of breast cancers, especially in the central portion of cancer where desmoplasia takes place [8]. The TCGA sample protocol mandated that samples be collected from the location within the bulk tumor that contains an average of 60% tumor cell nuclei with less than 20% necrosis [58]. Thus, it needs to be noted that the adipocytes analyzed in the current study are intratumoral adipocytes located closer to the center of the tumor. These are distinctively different from adipocytes from the peri-tumoral location, where adipocytes surrounding but not infiltrating the tumor may be included. Intratumoral adipocyte-low tumors enriched for G2M checkpoint gene set, which we recently reported to associate with aggressive characteristics in breast cancer [59]. The tumors with enhanced G2M checkpoint pathway demonstrated more advanced AJCC stage as well as more advanced Nottingham pathological grade, and thus increased risk of distant metastasis. We also reported that enhanced E2F signaling associated with clinically aggressive breast cancer [60]. Taken together, we speculate that the aggressive characteristics of adipocyte high tumors that cause inflammation and metastasis may be counterbalanced by the highly proliferative biology of adipocyte low tumors, leading to the no survival difference of intratumoral adipocyte-high versus low tumors.

This report is a retrospective study that mainly used publicly open databases, such as TCGA. There are several caveats to this study. TCGA and other publicly available breast cancer cohorts used in this study lack information on social history, which prevented us from analyzing the association between intra-tumoral adipocyte and social history, including smoking. One of the main strengths, as well as a limitation of this study, is that we transcriptomically defined the adipocyte. This approach allows us to assess the amount of intratumoral adipocytes in any bulk tumor with gene expression data, even when there was no intention to measure adipocytes at the time of the initial sample collection. However, because this approach utilizes the publicly available cohorts, our analyses are limited to the clinical parameters, the quality, and exact spatial location of where the sample was taken by the original authors. Further, although we have demonstrated that our methodology appropriately correlated with markers of mature adipocyte, there is a possibility that we may be picking up signals other than adipocytes. In addition, the mainstays of analyzing immune cells are immunohistochemistry and flow cytometry. However, these analyses were not performed due to the lack of access to the patient samples. We also did not conduct any in vitro and in vivo experiments, and thus are reliant on current literature to understand the underlying mechanisms.

In conclusion, a high amount of intratumoral adipocyte was associated with inflammation, metastatic pathways, cancer stemness, and favorable tumor immune microenvironment; whereas, a low amount of intratumoral adipocytes was associated with a highly proliferative tumor in ER-positive breast cancer. This cancer biology may explain the reason why patient survival did not differ by the amount of adipocytes.

## 4. Materials and Methods 

### 4.1. Data Acquisition

A total of 1090 patient clinicopathological and gene expression data of The Cancer Genome Atlas (TCGA) Pan-Cancer study (TCGA PanCancer Atlas) was obtained through cBioPortal as previously described [17,19,40,41,48,60,61].

Clinical and normalized microarray-based gene expression data were obtained from the following cohorts that underwent neoadjuvant chemotherapies; Symmans et al. (GSE25066; *n* = 508, taxane and anthracycline) [44], Noguchi et al. (GSE32646; *n* = 115; 5-FU, epirubicin, cyclophosphamide, and paclitaxel) [45], and Shi et al. (GSE20194; *n* = 278; paclitaxel, 5-FU, cyclophosphamide and doxorubicin) [46].

The data of a study by Yau et al. (*n* = 683) [62] was obtained through UCSC Xena (http://xena.ucsc.edu, University of California Santa Cruz, Santa Cruz, CA, USA) [63]. Given that all the cohorts used in this study were in a publicly available de-identified database, approval from the Institutional Review Board was waived for this study.

### 4.2. Gene Set Enrichment Analysis (GSEA)

Gene set enrichment analysis (GSEA) was a publicly available software (GSEA version 4.0) provided by the Broad Institute (Cambridge, MA, USA) (http://software.broadinstitute.org/gsea/index.jsp). For GSEA, we utilized Hallmark gene sets used for this study, as previously described [14,18,19,40,41,47,64,65]. False discovery rate (FDR) of 0.25 was utilized to define the statistical significance of GSEA throughout the study, as the Broad Institute recommends.

### 4.3. Immune Cell Composition and Scores Related with Immune Activity

Intratumoral adipocytes were defined with a computational algorithm, xCell, which utilized the transcriptome data as described in the journal, Genome Biology, in 2017 by Aran D et al. [42]. In addition, xCell was used to analyze the cell composition of all the other immune cells examined in this study. Cytolytic activity (CYT) was calculated using the geometric mean of granzyme A and Perforin 1 expression values, as described previously [39,59,66].

### 4.4. Statistical Analysis

All the statistical analysis was performed using R software (version 4.0.2; http:///www.r-project.org/). Kaplan–Meier survival analysis was performed with greyzoneSurv packages in R for the survival analysis. One-way ANOVA or Fisher’s exact test was used to determine the significance of differences for groups. Other box plots were analyzed with one-way ANOVA. A two-sided *p*-value  <  0.05 was considered statistically significant. All boxplots are of the Tukey type, and the boxes depicted medians and inter-quartile ranges.

## Figures and Tables

**Figure 1 ijms-21-05744-f001:**
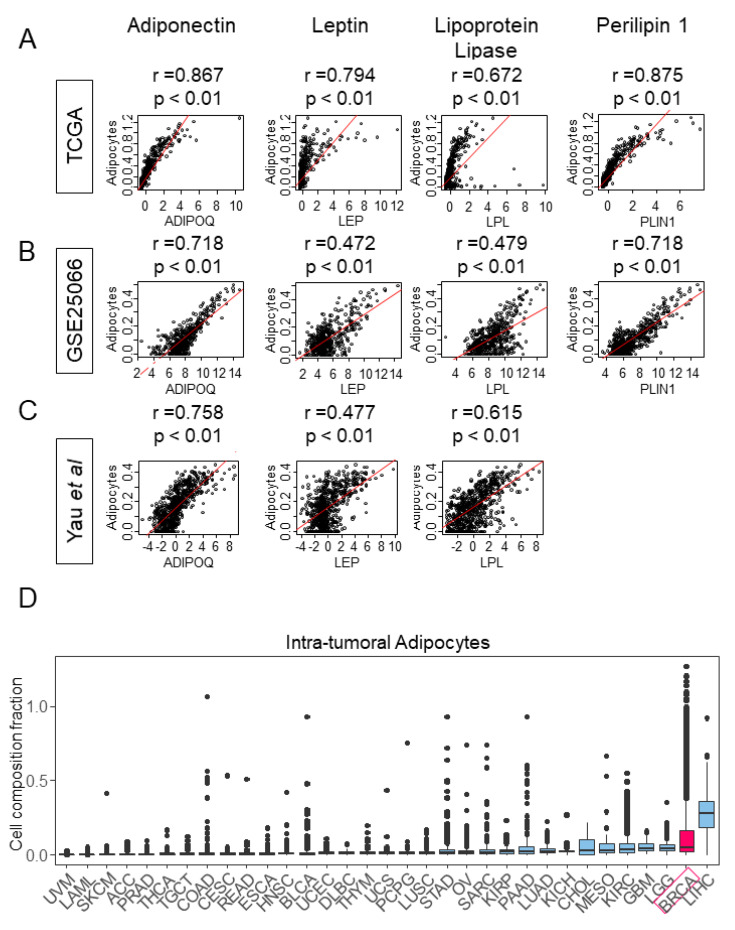
Transcriptionally defined adipocytes have a moderate to strong correlation with representative markers of mature adipocytes. The amount of intratumoral adipocytes in various cancers is also represented. (**A**) The Cancer Genome Atlas (TCGA) cohort (**B**) GSE25066 (**C**) Study by Yau et al. (**D**) The distribution of the amount of intratumoral adipocytes in various cancer types. ADIPOQ, adiponectin; LEP, leptin; LPL, lipoprotein lipase; PLIN1, perilipin 1; ACC, Adrenocortical carcinoma; BLCA, Bladder Urothelial Carcinoma; BRCA, Breast invasive carcinoma; CESC, Cervical squamous cell carcinoma and endocervical adenocarcinoma; CHOL, Cholangiocarcinoma; COAD, Colon adenocarcinoma; DLBC, Lymphoid Neoplasm Diffuse Large B-cell Lymphoma; ESCA, Esophageal carcinoma; GBM, Glioblastoma multiforme; HNSC, Head and Neck squamous cell carcinoma; KICH, Kidney Chromophobe; KIRC, Kidney renal clear cell carcinoma; KIRP, Kidney renal papillary cell carcinoma; LAML, Acute Myeloid Leukemia; LGG, Brain Lower Grade Glioma; LIHC, Liver hepatocellular carcinoma; LUAD, Lung adenocarcinoma; LUSC, Lung squamous cell carcinoma; MESO, Mesothelioma; OV, Ovarian serous cystadenocarcinoma; PAAD, Pancreatic adenocarcinoma; PCPG, Pheochromocytoma and Paraganglioma; PRAD, Prostate adenocarcinoma; READ, Rectum adenocarcinoma; SARC, Sarcoma; SKCM, Skin Cutaneous Melanoma; STAD, Stomach adenocarcinoma; TGCT, Testicular Germ Cell Tumors; THCA, Thyroid carcinoma; THYM, Thymoma; UCEC, Uterine Corpus Endometrial Carcinoma; UCS, Uterine Carcinosarcoma; UVM, Uveal Melanoma. *p*-value  <  0.05 was considered statistically significant. *r* represents a Spearman’s correlation coefficient.

**Figure 2 ijms-21-05744-f002:**
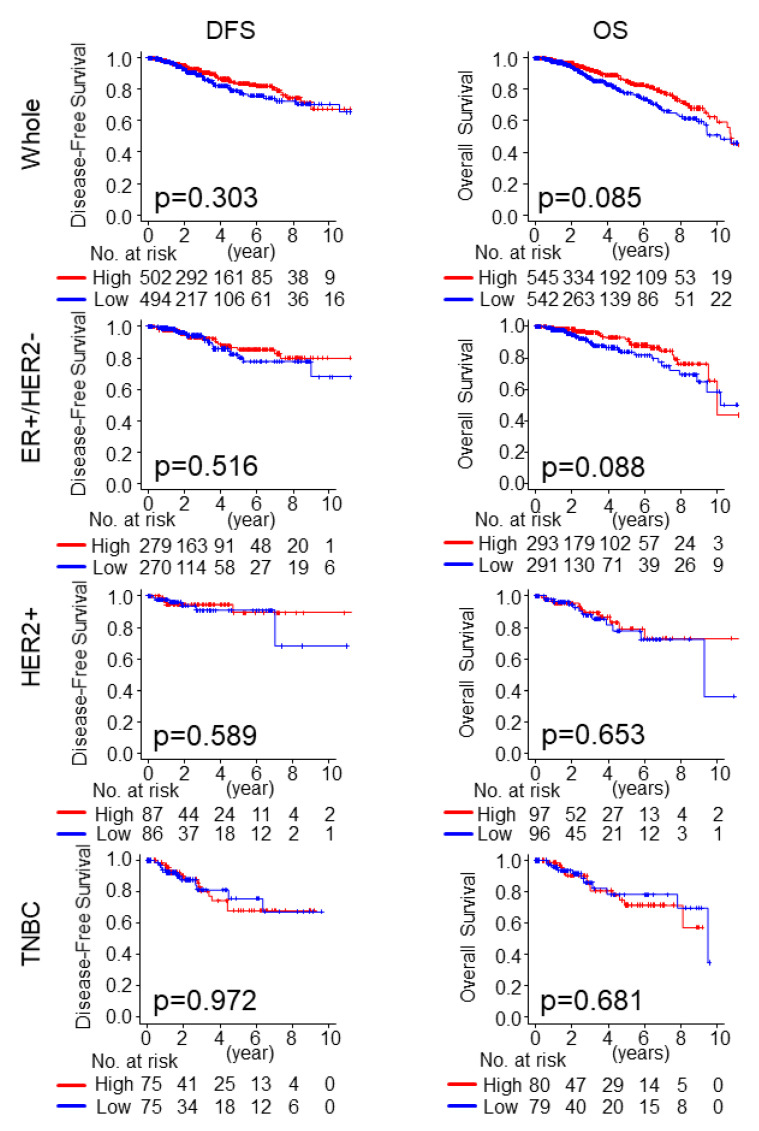
Kaplan Meier survival analysis (Disease-free survival (DFS) and overall survival (OS)) of adipocyte high (red line) vs. low (blue line) breast cancer in whole TCGA breast cancer cohort, ER+/HER2- subtype, Her2+ subtype, and TNBC subtype. The number of patients at risk is shown below the X-axis of each panel. DFS, disease-free survival; OS, overall survival; ER+/HER2-, ER-positive, and HER2 negative; HER2+, HER2 positive; TNBC, triple-negative breast cancer. *p*-value  <  0.05 was considered statistically significant.

**Figure 3 ijms-21-05744-f003:**
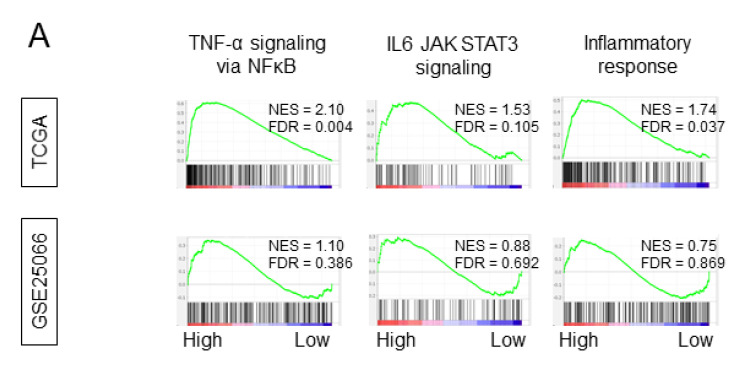
The gene set enrichment analysis (GSEA) of adipocyte high tumors in the whole cohort of TCGA breast cancer and GSE25066. (**A**) GSEA of Hallmark inflammation-related gene sets. (**B**) GSEA of Hallmark metastasis- and cancer stemness-related gene sets. (**C**) GSEA of Hallmark immune response-related gene sets. NES, normalized enrichment score; FDR, false discovery rate. FDR < 0.25 was considered statistically significant.

**Figure 4 ijms-21-05744-f004:**
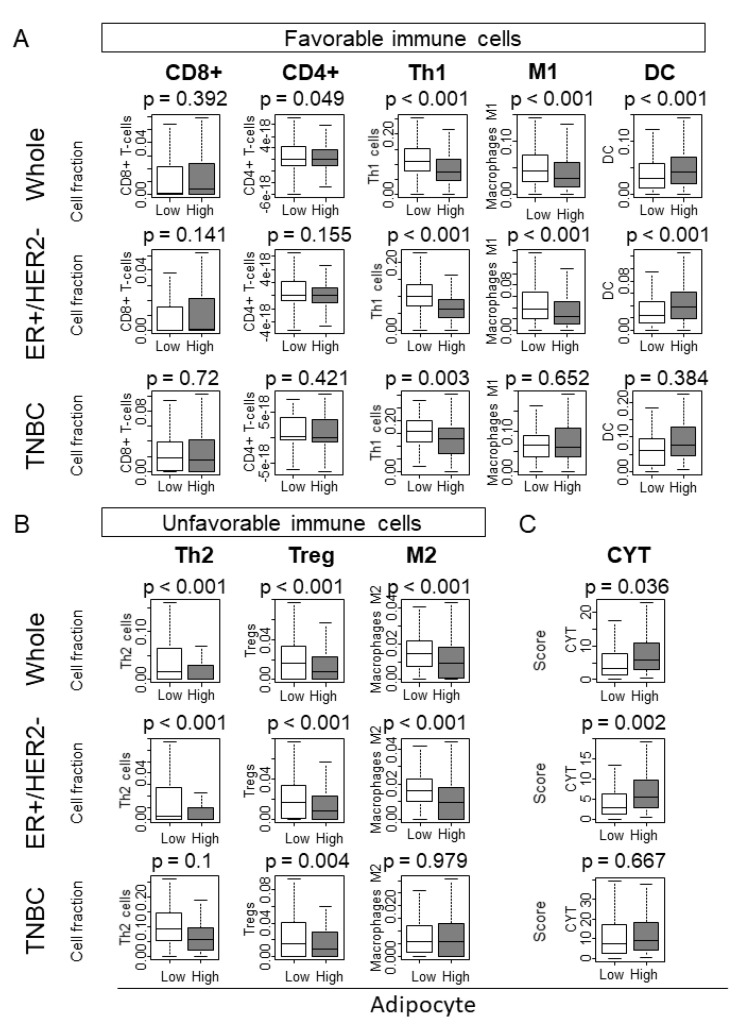
Immune cell composition by high and low adipocytes in the whole TCGA cohort, ER+/Her2-, and TNBC subtypes. (**A**) Favorable immune cells: CD8+ (CD8+ T cell), CD4+ (CD4+ T cell), Th1 (type 1 helper T cell), M1, (M1 macrophage), DC (Dendritic Cell). (**B**) Unfavorable immune cells: Th2 (type 2 helper T cell), Treg (regulatory T cell), M2 (M2 macrophage. (**C**) Cytolytic activity score (CYT). ER+/HER2-, ER-positive and HER2 negative; HER2+, HER2 positive; TNBC, triple-negative breast cancer. *p*-value  <  0.05 was considered statistically significant.

**Figure 5 ijms-21-05744-f005:**
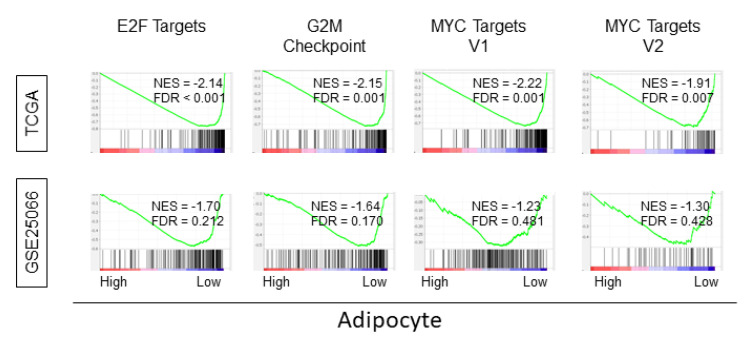
GSEA of adipocyte low tumors enriched Hallmark cell cycle and cell proliferation-related gene sets in whole TCGA cohort. NES, normalized enrichment score; FDR, false discovery rate. FDR < 0.25 was considered statistically significant.

**Figure 6 ijms-21-05744-f006:**
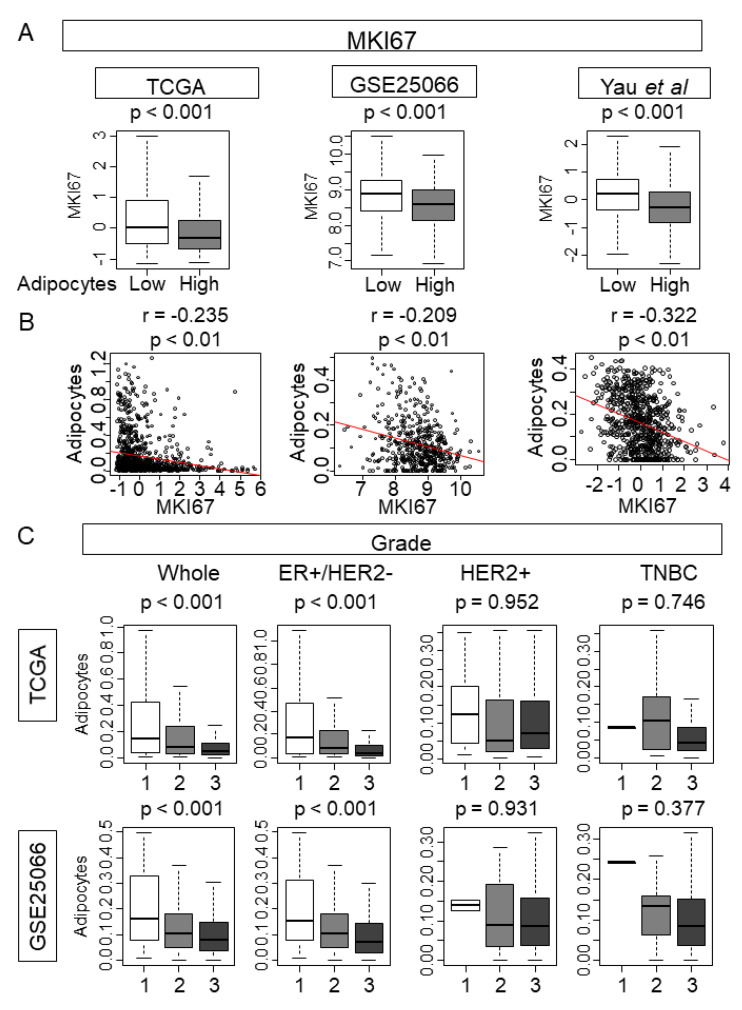
The amount of intratumoral adipocytes correlate with *MKI67* expression and Nottingham pathological grade. (**A**) *MKI67* gene expression by intratumoral adipocyte High (shaded box) vs. Low (open box) in whole TCGA, GSE25066, and Yau et al.’s cohort. (**B**) Pearson correlation curve of *MKI67* expression and adipocyte amount in the whole TCGA, GSE25066 and Yau et al.’s cohort. (**C**) Nottingham pathological grade (grade 1: Open box, grade 2: Shaded box, grade 3: Closed box) and amount of adipocytes in whole cohort and ER+/HER2-, Her2+, TNBC subtype in TCGA and GSE25066 cohort. ER+/HER2-, ER-positive and HER2 negative; HER2+, HER2 positive; TNBC, triple-negative breast cancer. *p*-value  <  0.05 was considered statistically significant. *r* represents a Spearman’s correlation coefficient.

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
