# Peer review of "Intratumoral Adipocyte-High Breast Cancer Enrich for Metastatic and Inflammation-Related Pathways but Associated with Less Cancer Cell Proliferation"

_ijms, 2020, doi:10.3390/ijms21165744_

Round 1

Reviewer 1 Report

The current manuscript aimed to investigate the relation between adipocytes and metastatic and inflammation-related pathways in breast cancer. The authors could successfully address their thoughts and hypotheses by a well-designed study. The results here demonstrated a reason why patient survival did not differ by the amount of adipocytes. I believe that this study offers some valuable findings to the readers who are interested in different aspects of breast cancer and its metastatic behavior.

Sincerely yours

Reviewer 2 Report

This is an interesting study on adipocytes in breast cancer tumors. The hypothesis of the study has been falsified: high content of adipocytes in breast cancer tumors are correlated with reduced patient survival. It appeared that survival of patients was not affected by the amount of adipocytes in the tumors in an in silico analysis using 3 data bases.

1. The study has been performed properly and the use of the English language is fine but there are too many unnecessary errors in the text.

A few examples: lines 159 and 286: '(enter reference)'.

line 73: 'the favorable tumor immune which leads...'

line 164: 'and cancer stemness and are also enriched'

Furthermore, unnecessary capitals are used throught the text.

2. Self-citation is abundant and, in my opinion, not acceptable: 32 references out of a total of 50 (64%!!) are self-citations.

3. One of the 3 data bases is the one of Yau et al (line 94). However this reference does not appear in the reference list. In M&M (line 333) another reference is given (39) without any further explanation.

Reviewer 3 Report

The manuscript entitled “Intratumoral adipocyte-high breast cancer enrich for metastatic and inflammation-related pathways, but associated with less cancer cell proliferation.” is an intrigue study about cancer-associated adipocytes are known to cause inflammation and lead to the cancer progression, and metastasis.
This manuscript is well written and balance, so far it should be published after some minor revisions.
The authors showed that high amount of intratumoral adipocyte in breast cancer was associated with inflammation, metastatic pathways, cancer stemness and favorable tumor immune microenvironment. However, low amount of adipocytes wasassociated with highly proliferative tumor in ER-positive breast cancer.
Please, complete the data (if possible): did the number of addipocytes inside the tumor influence the treatment of breast cancer?
Smoking is a known risk factor for both obesity and breast cancer.
Please fill in the data on how smoking influences the development of the number of addipocytes.

The quality of the figures should be improved as they are not legible.
